# In-Depth Study on the Effects of Impurity Ions in Saline Wastewater Electrolysis

**DOI:** 10.3390/molecules28124576

**Published:** 2023-06-06

**Authors:** Qicheng Pan, Peixuan Zhao, Linxia Gao, Huimin Liu, Hongyun Hu, Lu Dong

**Affiliations:** 1College of Resources and Environment, Hubei University of Technology, Wuhan 430068, China; 2State Key Laboratory of Coal Combustion, School of Energy and Power Engineering, Huazhong University of Science and Technology, Wuhan 430074, Chinahongyunhu@hust.edu.cn (H.H.); ludong@hust.edu.cn (L.D.); 3Shenzhen Research Institute of Huazhong University of Science and Technology, Shenzhen 518063, China

**Keywords:** saline wastewater, electrolysis, dechlorination, impurity ions

## Abstract

Concentration followed by electrolysis is one of the most promising ways for saline wastewater treatment, since it could produce H_2_, Cl_2,_ and an alkaline solution with deacidification potential. However, due to the diversity and difference of wastewater, knowledge on the suitable salt concentration for wastewater electrolysis and the effects of mixed ions are still lacking. In this work, electrolysis experiments of mixed saline water were conducted. The salt concentration for stable dechlorination was explored, with in-depth discussions on the effects of typical ions such as K^+^, Ca^2+^, Mg^2+^, and SO_4_^2−^. Results showed that K^+^ had a positive effect on the H_2_/Cl_2_ production of saline wastewater through accelerating the mass transfer efficiency in the electrolyte. However, the existence of Ca^2+^ and Mg^2+^ had negative effects on the electrolysis performance by forming precipitates, which would adhere to the membrane, reduce the membrane permeability, occupy the active sites on the cathode surface, and also increase the transport resistance of the electrons in the electrolyte. Compared to Mg^2+^, the damaging effect of Ca^2+^ on the membrane was even worse. Additionally, the existence of SO_4_^2−^ reduced the current density of the salt solution by affecting the anodic reaction while having less of an effect on the membrane. Overall, Ca^2+^ ≤ 0.01 mol/L, Mg^2+^ ≤ 0.1 mol/L and SO_4_^2−^ ≤ 0.01 mol/L were allowable to ensure the continuous and stable dechlorination electrolysis of saline wastewater.

## 1. Introduction

With the high development of economy and the continuous expansion of industrial scale, the amount of saline wastewater—which is usually rich in inorganic salts (Na^+^, Cl^−^, K^+^, SO_4_^2−^, Ca^2+^, Mg^2+^)—generated from chemical industry, printing and dyeing, and seawater utilization in China increases year by year. Inappropriate disposal would cause serious environmental problems, such as soil hardening, eutrophication of water, and waste of salt and water resources [1,2]. Evaporation crystallization technology is a mature and recognized process for salt-containing wastewater to achieve “zero discharge” [3]. However, the disposal of the mixed salt is a big problem that needs to be solved [4]. With the wide application of alkaline water electrolysis [5,6,7], advanced electrochemical technology [8], brine electrolysis technology in seawater desalination, chlor-alkali industry, and wastewater disposal [9], concentration followed by electrolysis becomes one of the most promising ways for saline wastewater treatment [10], as it could produce H_2_, Cl_2,_ and an alkaline solution with deacidification potential. Benefited by the advantages of the high purity of separated gas, ion exchange membrane electrolysis [11,12] is now the most common used technology in the brine electrolysis process, even though its electrolytic efficiency depends largely on the performance of the ion membrane. Since there are all kinds of ions with various concentrations in saline wastewater, the pores of the ion membrane might be blocked, resulting in higher tank voltage and lower electrolysis efficiency [13,14]. Therefore, it is necessary to explore the suitable concentrations of impurity ions during saline wastewater electrolysis to ensure a stable and economic electrolytic performance with minimal damage to the ion membrane.

Numerous studies [15,16] have studied the hydrogen production performance of alkaline water electrolysis, especially the effect of +1 valent cations such as K^+^ and Na^+^ on hydrogen production. Fariba et al. [17] pointed out that KOH was more effective than NaOH as an electrolyte during water electrolysis. The findings of Liu et al. [18] illustrated that caused by the better mass transfer capacity of K^+^, the efficiency of electrolysis with KOH was increased by about 10% compared to the same concentration of NaOH. Sun et al. [19] found that the electrolysis voltage of a 1 mol/L KOH solution for overall seawater splitting was much lower than a 1 mol/L NaOH solution. As to the effects of +2/−2 valence ions such as Ca^2+^, Mg^2+^, and SO_4_^2−^, the studies of Wang et al. [20] showed that after the +2/−2 valence ions entered the membrane, the crystals formed by the OH^−^ reverse osmosis in the membrane would block the channel in the membrane, causing a reduction of the exchange ability of the ion membrane. As a result, the voltage of the ion membrane tank increased with a sharp decline in the current efficiency. Furthermore, Khajouei et al. [21] and Sun et al. [22] found that aluminum could be dissolved into colloidal aluminum in acidic brine and then reacted with SiO_2_ to form an aluminum silicate precipitation, which was also a threat to the ion membrane. Furthermore, the impure silicon compounds could absorb and constantly react with other impurities to form more precipitations. It indicated that mixed impurity ions may cause even worse effects than single kinds of impurity ions. Note that these studies were all based on pure water electrolysis. However, Buddhi et al. [23] reported that the electrolysis efficiency of the NaCl solution was better with trace amounts of Mg^2+^ in it, demonstrating the complexity of the ion effects under different electrolyte environments. In fact, the impurity ions were usually limited to very low concentrations to ensure the H_2_/Cl_2_ production efficiency for either water electrolysis or saturated NaCl solution electrolysis [24,25,26]. In detail, the concentration of SO_4_^2−^ should be no higher than 0.05 mol/L for the purpose of 30% NaOH solution production and much lower for 42% or 50% NaOH solution production [27] in the chlor-alkali industry. Furthermore, the mass fraction of total Ca^2+^ and Mg^2+^ in the secondary refined brine ($w_{\left(Ca^{2+}+Mg^{2+}\right)}$) was commonly used as the system index in saturated NaCl solution electrolysis [28,29]. A value less than 10^−6^ mol/L was required to ensure electrolysis efficiency and liquid alkali purity.

Overall, previous studies cared more about the electrolysis efficiency and yield. Very strict limits on ion concentration were put forward in the saltwater electrolysis process, leading to a high impurity removal cost and a large increase in the electrolytic cost [30,31,32]. Considering the much higher concentrations and various kinds of impurities in saline wastewater compared to saturated salt solutions (in the chlor-alkali industry), it was uneconomic to pay the extremely high impurity removal cost for the purpose of saline wastewater desalination through electrolysis. In another way, the allowable concentration range of the impurity ions might become wider if the purity of the product was not pursued. For example, the alkaline solutions produced by electrolysis can be used to remove acid gases from power plants or incinerators; in these situations, the quality of the alkaline was not strictly demanded. In this case, the saline wastewater was expected to be transferred into gas products (H_2_ and Cl_2_ yield might not be so high) and alkaline products (NaOH might not be so pure) simultaneously, while in a more economical way.

Following this line of thought, systematic electrolysis experiments were conducted in the work, with detailed discussion on the salt concentration for stable dechlorination, the allowable ion species, concentrations for economic saltwater electrolysis, as well as the pollution mechanisms of impurity ions on the ion membrane. Findings in the work are aimed to provide reference on the dichlorination and resource utilization of saline wastewater in future.

## 2. Results and Discussion

### 2.1. Effect of NaCl Concentration

To investigate the most suitable salt concentration for stable H_2_/Cl_2_ rather than H_2_/O_2_ production, a wide range of the NaCl mol ratio in saltwater from 2–4.5 mol/L was tested. The effects of salt concentration on the current density and pH of the cathode solution are shown in Figure 1, where in Figure 1a, the real-time current density was monitored during the whole electrolysis process. Figure 1b lists the change of pH before and after electrolysis, with the average current density marked by a red line.

It is seen in Figure 1a that for each NaCl concentration case, the current density of the electrolysis system increased at an early stage. It was mainly due to how H^+^ needed to be adhered to the active site of the electrode to get electrons. However, when two H^+^ ions were adhered to the electrode, the generated H_2_ would complete the hydrogen evolution reaction. The duration time to reach equilibrium of the hydrogen adhesion-reaction-evolution process was actually the period when the current density increased gradually. When the NaCl mass concentration was lower than 4 mol/L, the current density tended to be stable after 200 min of continuous electrolysis. As the NaCl mass concentration continued to increase (up to 4.25 mol/L or 4.5 mol/L), the current density increased first and then decreased for a long time afterwards. The biggest explanation for this phenomenon was that with too many Na^+^ ions in the electrolyte solution, the Na^+^ ions would be strongly pulled by the electrode and accumulate near the cathode, which would prevent H^+^ from finding active sites on the cathode. At the same time, the Na^+^ ions in the anode cell tended to have a concentration polarization effect near the cation exchange membrane [33]. It further increased the tank voltage in the electrolytic cell, and thus the current density was reduced.

For every H_2_ molecule produced, 2OH^−^ ions were generated which increased the pH of the cathode solution, and the pH changed in the same way as the current density, as seen in Figure 1b. It is concluded that compared to other salt concentration cases, a 4 mol/L salt concentration showed the best performance with the current density close to 340 mA/cm^2^, and the pH exceeded 13.9 after electrolysis. In this case, without special explanation, a 4 mol/L salt concentration was applied in the following discussion sections on the effects of impurity ions.

### 2.2. Effect of K^+^

The effect of K^+^ was first discussed with 4 mol/L Cl^−^ as a basis. As the concentration of K^+^ in the electrolyte increased from 0 to 2 mol/L (equal to that of Na^+^ in the electrolyte solution), the current density became much higher, from 340 mA/cm^2^ to 480 mA/cm^2^, as shown in Figure 2. The acceleration of K^+^ was mainly due to how K^+^ had an extra electron layer than Na^+^, which was more electronegative and would be more easily attracted by the cathode. Thus, K^+^ moved faster from anode to cathode in the electrolyte, and the tank voltage was smaller, which increased the current density of the electrolysis system.

It was also observed in Figure 2 that there was a jump in the current density when the concentration of K^+^ was higher than 1 mol/L. However, as the concentration of K^+^ further increased from 1.5 to 2 mol/L, little change in the current density was observed. This suggested that 1.5 mol/L was the appropriate K^+^ concentration for electrolysis. Though there is no clear explanation for the sudden rise from 1 to 1.5 mol/L at present, the little change at K^+^ > 1.5 mol/L was probably due to how the high concentration of K^+^ reduced the permeability of the ionic membrane. Additionally, the relationship between the H_2_/Cl_2_ yield (*y*) and the current density (*x*) was analyzed and plotted as shown in Figure 3. In detail, for H_2_: *y* = 0.0165*x* − 0.5317 and for Cl_2_: *y* = 0.0163*x* − 0.5104. The very close slope of Cl_2_ and H_2_ indicated that dechlorination can be achieved through wastewater electrolysis. Meanwhile, the slope of Cl_2_ production was a little bit lower compared to H_2_, which was mainly due to the fact that a trace of the O_2_ evolving reaction also occurred at the anode at the same time.

### 2.3. Effect of Ca^2+^ and Mg^2+^

With 4 mol/L Cl^−^ as a basis (the cations were composed of Na^+^ and Ca^2+^/Mg^2+^), the effects of +2 cations Ca^2+^ (set from 0 to 0.1 mol/L) and Mg^2+^ (set from 0 to 0.2 mol/L) on the current density and the electrolyte pH were also tested as seen in Figure 4. In the cases of Ca^2+^ < 0.01 mol/L (see Figure 4a) and Mg^2+^ < 0.1 mol/L (see Figure 4c), the current density increased and tended to be stable with increasing electrolysis time. For Ca^2+^ doping cases, the current density decreased slightly from 340 mA/cm^2^ with 0 mol/L Ca^2+^ to 320 mA/cm^2^ with 0.01 mol/L Ca^2+^. For Mg^2+^ doping cases, the current density rarely changed when the concentration of Mg^2+^ increased from 0 mol/L to 0.05 mol/L. As the Mg^2+^ increased to 0.1 mol/L, the current density became lower in the early stages (<100 min), then increased up to 400 mA/cm^2^ after 210 min of electrolysis. The even-higher current density with 0.1 mol/L Mg^2+^ in the electrolyte might because the enhanced mass transfer when part of Mg^2+^ was transferring from the anode side to the cathode side. Overall, the negative effects of +2 cations were not apparent if Ca^2+^ was less than 0.01 mol/L and Mg^2+^ was less than 0.1 mol/L.

However, when the concentration of Ca^2+^ and Mg^2+^ became higher, the current density decreased sharply and then increased to reach a new stable current density with electrolysis time (Figure 4a,c). For example, the current density decreased from 340 mA/cm^2^ without Ca^2+^ doping to ~210 mA/cm^2^ with 0.02 mol/L Ca^2+^ doping after 90 minutes’ electrolysis; an over 38% decline was observed. When the concentration of Ca^2+^ further increased to 0.1 mol/L, the decreasing tendency of the current density lasted longer to 180 min, while the lowest current density was similar to that of the 0.02 mol/L Ca^2+^ case. As for the effect of Mg^2+^, the current density decreased to 225 mA/cm^2^ with 0.15 mol/L Mg^2+^ doping, and only 160 mA/cm^2^ with 0.2 mol/L Mg^2+^ doping. At the same time, the decreasing tendency of the current density lasted longer, from 60 min to 120 min, with higher Mg^2+^ doping. In fact, a very similar phenomenon happened on Ca^2+^ and Mg^2+^, and was caused by the precipitation of hydroxide generated by Ca^2+^/Mg^2+^ and OH^−^. The Ca(OH)_2_ and Mg(OH)_2_ precipitates would adhere to the membrane, which reduced the membrane permeability and thus increased the tank voltage. The precipitates could also occupy the active sites on the cathode surface, which reduced the electrode reaction rate and led to a slower electron transfer rate. In addition, the hydroxides that were suspended in the electrolyte would increase the transport resistance of the electrons, also resulting in a higher tank voltage. As the reaction continued, the thickness of the precipitates on the surface of the membrane increased and partly fell to the bottom of the electrolytic cell due to gravity. That is why the current density with Ca^2+^ > 0.01 mol/L or Mg^2+^ > 0.1 mol/L showed a rising trend after several minutes while keeping lower than the cases with lower concentrations of Ca^2+^ or Mg^2+^.

Furthermore, the pH values of the electrolyte in the cathode cell before and after electrolysis were tested with Ca^2+^/Mg^2+^ doping, as shown in Figure 4b,d. It was seen that the electrolyte showed strong basicity after electrolysis, which had good deacidification potential. Combined with the electrolysis performance under various Ca^2+^ and Mg^2+^ concentrations, Mg^2+^ and Ca^2+^ in the saline wastewater are suggested to be no higher than <0.1 mol/L and <0.01 mol/L, respectively, from an engineering application perspective (economic and stable).

In addition, the relationships between the H_2_/Cl_2_ yield (*y*) and the current density (*x*) for Ca^2+^/Mg^2+^ doping cases were linear fitted as seen in Figure 5. For Ca^2+^, the H_2_ yield could be described by $y_{H_{2}}$ = 0.0151*x* − 0.0181, and the Cl_2_ yield could be described by $y_{{\mathrm{Cl}}_{2}}$ = 0.0143*x* − 0.1138. For Mg^2+^, the H_2_ yield was $y_{H_{2}}$ = 0.0149*x* + 0.0586, and the Cl_2_ yield was $y_{{\mathrm{Cl}}_{2}}$ = 0.0149*x* − 0.0381. The obvious difference between the slope of H_2_ and Cl_2_ in Ca^2+^ doping cases demonstrated that the competition of O_2_ production and Cl_2_ production was enhanced.

At the end of electrolysis, tight precipitation layers on the cathode side of the membrane which were difficult to remove by sweeping were observed for Ca doping cases. However, for Mg doping cases, sticky precipitation layers on the anode side of the membrane were observed. The SEM images and EDS analyses of the membrane surface before and after sweeping precipitations are shown in Figure 6. The big difference could be contributed to the different precipitation mechanisms of Ca(OH)_2_ and Mg(OH)_2_. At room temperature, the solubility product of Ca(OH)_2_ was 5.02 × 10^−6^ and the pH value for precipitation was around 11. Since the pH of the cathode chamber was higher than the anode chamber, the Ca^2+^ in the anode chamber passed through the membrane and precipitated on the cathode side. As for Mg, the solubility product of Mg(OH)_2_ was 5.61 × 10^−12^, and the pH value for precipitation was around 8. As the electrolysis proceeded, more and more OH^−^ generated in the cathode side, and some would pass through the cation exchange membrane by osmosis. The Mg(OH)_2_ precipitation tended to be generated once Mg^2+^ made contact with OH^−^ at the anode chamber with lower pH.

### 2.4. Effect of SO_4_^2−^

With 4 mol/L Na^+^ as a basis (anions were composed of Cl^−^ and SO_4_^2−^), the effect of SO_4_^2−^ on the current density as well as the electrolyte pH was analyzed and shown in Figure 7. For all the cases with SO_4_^2−^ doping, the current density increased gradually and then tended to stabilize with time. Compared to the electrolysis without SO_4_^2−^, the current density decreased to 270 mA/cm^2^ when the concentration of SO_4_^2−^ was set to 0.5 mol/L. It was mainly due to how SO_4_^2−^ co-existed in the NaCl solution; the diffusion coefficient of the NaCl electrolyte was reduced along with the lower transformation of Na^+^ or Cl^−^. Meanwhile, the competing oxygen evolution reaction in the anode solution was enhanced, in which four electrons were required [34]. Caused by the lower transmission rate of four electrons than a single electron, the electrode reaction rate in the anode cell decreased, same as the current density. On the other hand, since a cation exchange membrane was used in the experiments, SO_4_^2−^ could not pass through the membrane nor transfer charge, thus the current density did not change much when the concentration of SO_4_^2−^ ranged from 0.01 mol/L to 0.1 mol/L. However, the negative effect of SO_4_^2−^ became more significant when SO4^2−^ further increased, which was mainly contributing to the white precipitate (Na_2_SO_4_ of low solubility) generated on the membrane surface. It was also found that the curves in Figure 7a did not fall and rise as the 2+ cations (Ca^2+^ or Mg^2+^) did, mainly due to how there was no precipitation shed in the process.

Despite the white precipitate attached to the cation exchange membrane, the property of the membrane could be recovered well after simple water washing. It suggested that the existence of SO_4_^2−^ reduced the current density of the saline water through affecting the anodic reaction while having less of an effect on the membrane. The pH values of the electrolyte in the cathode cell before and after electrolysis were also tested with SO4^2−^ doping. Strong basicity (13.8–14.0) with deacidification potential was verified as seen in Figure 7b.

### 2.5. Electrochemical Analysis and Comparision of All Cases

The LSV curves and Tafel slopes of the anode for all cases are tested and shown in Figure 8 and Figure 9, respectively. The current efficiency of each case could be found in Appendix A. From Figure 8, one can see that the current density was almost 0 in the range of 0–1 V_Hg/HgO_, and then increased sharply, suggesting that saline water electrolysis started after 1 V_Hg/HgO_. For the specific cations at the same current density (mA/cm^2^), the potentials of the saline water became obviously lower with Na^+^, K^+^, and Mg^2+^ addition, or with ≤0.05 mol/L Ca^2+^ addition. Inversely, the appearance of SO_4_^2−^ increased the potentials of the saline water, and even 0.05 mol/L SO_4_^2−^ would cause devastating effects. In Figure 9, the Tafel slope of the 2 mol/L NaCl + 2 mol/L KCl system was the lowest (46 mV dec^−1^), followed by the 3.8 mol/L NaCl + 0.1 mol/L MgCl_2_ system (51 mV dec^−1^). The 3 mol/L NaCl^+^ 0.5 mol/L Na_2_SO_4_ system was the highest (105 mV dec^−1^). In addition, the Tafel slope of the Na-Ca-Cl system reached the lowest value with 0.05 mol/L Ca^2+^ addition (57 mV dec^−1^).

To ensure continuous and stable dichlorination electrolysis of the saline wastewater, both the electrochemical analysis (from the theoretical perspective) and the experiments conducted above (from the engineering application perspective) should be taken into account. Therefore, it was suggested that SO_4_^2−^ should be strictly limited. The concentration of Ca^2+^ and Mg^2+^ should be no higher than 0.01 mol/L and 0.1 mol/L, respectively.

## 3. Materials and Methods

Simulated saline wastewater was used, and the compositions of saline water solution with impurity ions were listed in Table 1. The total ionic charge (Na^+^ as a basis when discussing the effect of anions or Cl^−^ as a basis when discussing the effect of cations) was set to be 4 mol/L, a value to ensure constant Cl_2_ rather than O_2_ production in the anode. The reagents used in the experiment (NaCl, Na_2_SO_4_, KCl, CaCl_2_, and MgCl_2_·6H_2_O) were all analytically pure.

The schematic diagram of the experimental equipment is shown in Figure 10. The effective area of each electrode plate was 2 cm^2^, and the volume of the electrolyte in each chamber was 150 mL. The cathode plate was made of Q235 carbon steel while the anode plate was a DSA plate (titanium plate coated with IrO_2_-RuO_2_). The cation exchange membrane was a perfluoro-sulfonate ion-exchanged membrane (Nafion N117), manufactured by Dupont, Wilmington, DE, USA. The electrolysis experiments were conducted under a constant voltage of 12 V to ensure the current density was not lower than 200 mA/cm^2^. The very high voltage compared to some seawater electrolysis studies (less than 4 V) [35,36] was mainly caused by the large space between two electrodes and the high resistance of electron motion in the wastewater [37]. In each electrolysis experiment case, the first 30 min was reserved for system stabilization, along with continuous H_2_/Cl_2_ collection for the following 210 min using the drainage method. Considering the competitive evolution reaction of O_2_ and Cl_2_ at the anode side, the gas produced on the anode was first washed by a NaOH adsorption bottle to remove Cl_2_ before being collected. After each experiment, the ion exchange membrane was washed and dried at 50 °C in the oven for 0.5 h. Note that the blocked or broken membrane was replaced in time.

The whole electrolytic reaction process consisted of three steps: electrode reaction (anodic oxidation and cathodic reduction), migration of metal cations (Na^+^, K^+^, Ca^2+^ and Mg^2+^), and the precipitation of calcium/magnesium hydroxide. The electrode reactions occurring in the anode and cathode chambers were the chlorine and hydrogen evolution reactions, respectively. At the same time, the cations migrated from the anode to the cathode under the action of the electric field. Then, the Ca^2+^/Mg^2+^ combined with OH^−^ to form a Ca(OH)_2_/Mg(OH)_2_ precipitate. The reactions of the whole process were represented by Equations (1)–(3).
Anodic oxidation reaction: 2Cl^−^ − 2e^−^→Cl_2_↑(1)
Cathodic reduction reaction: 2H^+^ + 2e^−^→H_2_↑(2)
Precipitation reaction: M^2+^ + 2OH^−^→M(OH)_2_↓(M→Ca/Mg)(3)

For the electrochemical analyses, the reference electrode was Hg/HgO. The two-channel electrochemical workstation Donghua 7003 (DH7003) was used in this work. The linear volumetric sweep (LSV) was set as 0–2 V_Hg/HgO_ with 20 mV/s. The Tafel slope analysis condition was 0.1 mV/s, from 0 to 2 V_Hg/HgO_. The microscopic morphology and residual compositions of the ion exchange membranes were analyzed with a scanning electron microscope (Sigma 300, Carl Zeiss, Jena, Germany) and an X-ray energy spectrometer (Aztec M-Max 80, Oxford, UK). The cations (Na^+^, K^+^, Ca^2+^ and Mg^2+^) in the solution were determined by a microwave plasma atom emission spectrometer (MP-AES 4200, Agilent Technologies, Santa Clara, CA, USA). Anions were analyzed by an ion chromatograph (IC-2010, Tosoh, Japan). The pH in the salt solution was measured by a pH meter (MP-PHSJ-4F, Shanghai Inesa, China).

## 4. Conclusions

(1) The current density of saline water electrolysis increased first, and then decreased with increasing NaCl concentration. A 4 mol/L NaCl concentration showed the best performance, with the current density close to 340 mA/cm^2^ and pH exceeding 13.9 after electrolysis.

(2) Replacement of Na^+^ with K^+^ had a positive effect on the H_2_/Cl_2_ production through accelerating the mass transfer efficiency in the electrolyte. When the concentration of K^+^ was higher than 1 mol/L, the acceleration of K^+^ became less significant.

(3) Ca^2+^ and Mg^2+^ had negative effects on the electrolysis performance by forming precipitates on the membrane surface or suspension in the electrolyte. SO_4_^2−^ reduced the current density of the salt solution by affecting the anodic reaction while having less of an effect on the membrane. The allowable concentration of +2/−2 valence impurity ions were Ca^2+^ ≤ 0.01 mol/L, Mg^2+^ ≤ 0.1 mol/L and SO_4_^2−^ ≤ 0.01 mol/L.

## Figures and Tables

**Figure 1 molecules-28-04576-f001:**
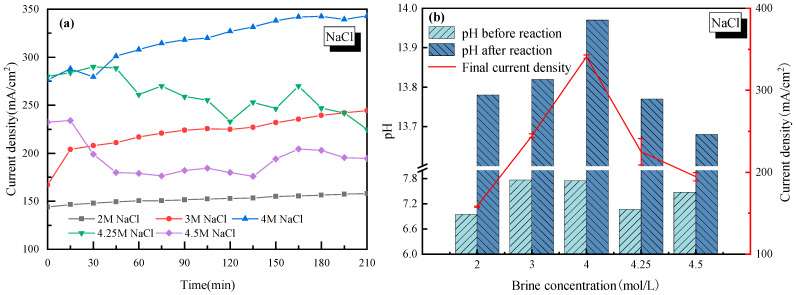
Effect of NaCl concentration on the electrolysis performance of saline water. (**a**) Current density vs. time; (**b**) pH before and after electrolysis.

**Figure 2 molecules-28-04576-f002:**
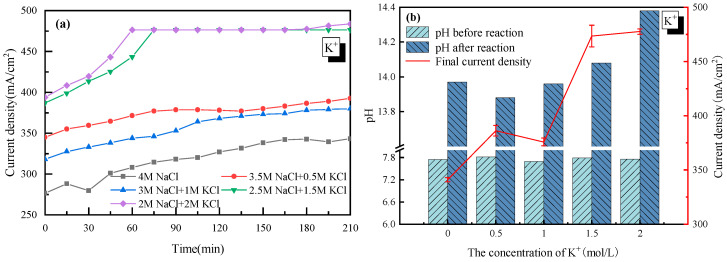
Effect of K^+^ on the electrolysis performance of saline water. (**a**) Current density vs. time; (**b**) pH before and after electrolysis.

**Figure 3 molecules-28-04576-f003:**
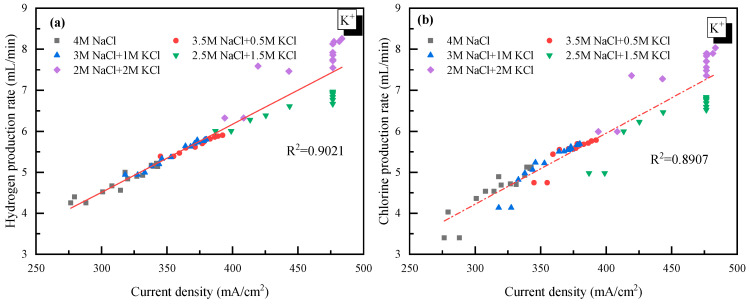
Relationship between the gas yield and current density with K^+^ doping. (**a**) H_2_ yield vs. current density; (**b**) Cl_2_ yield vs. current density.

**Figure 4 molecules-28-04576-f004:**
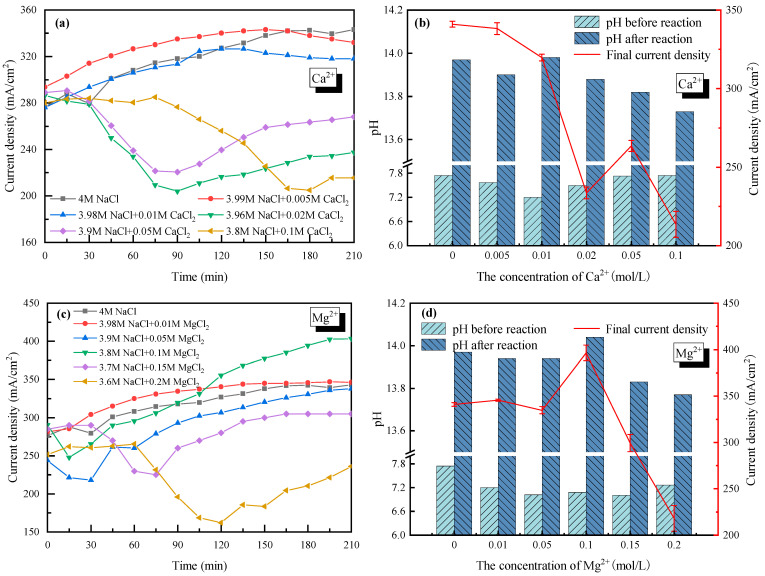
Effect of Ca^2+^ and Mg^2+^ on the electrolysis performance of saline water. (**a**) Current density vs. time with Ca^2+^ doping; (**b**) pH before and after electrolysis with Ca^2+^ doping; (**c**) Current density vs. time with Mg^2+^ doping; (**d**) pH before and after electrolysis with Mg^+^ doping.

**Figure 5 molecules-28-04576-f005:**
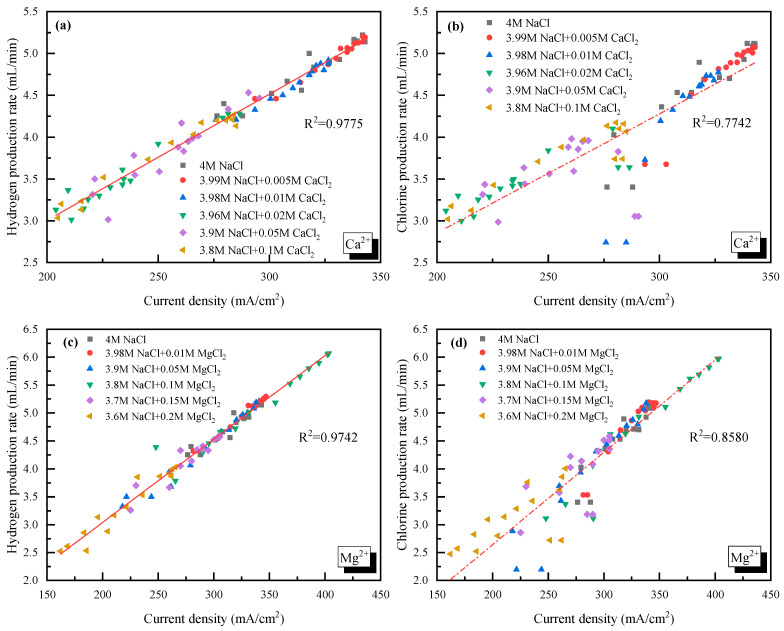
Relationship between the gas yield and current density with Ca^2+^ or Mg^2+^ doping. (**a**) H_2_ yield vs. current density with Ca^2+^ doping; (**b**) Cl_2_ yield vs. current density with Ca^2+^ doping; (**c**) H_2_ yield vs. current density with Mg^2+^ doping; (**d**) Cl_2_ yield vs. current density with Mg^2+^ doping.

**Figure 6 molecules-28-04576-f006:**
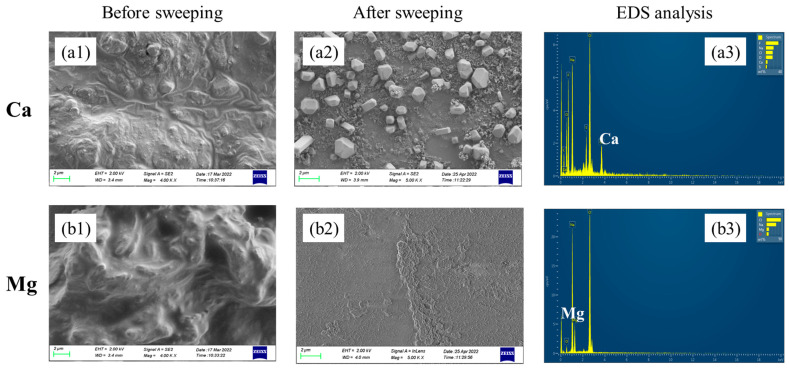
SEM images of the membrane surface before and after precipitation sweeping. (**a1**) Ca^2+^ effect before sweeping; (**a2**) Ca^2+^ effect after sweeping; (**a3**) EDS analysis of the Ca-containing membrane after sweeping; (**b1**) Mg^2+^ effect before sweeping; (**b2**) Mg^2+^ effect after sweeping; (**b3**) EDS analysis of the Mg-containing membrane after sweeping.

**Figure 7 molecules-28-04576-f007:**
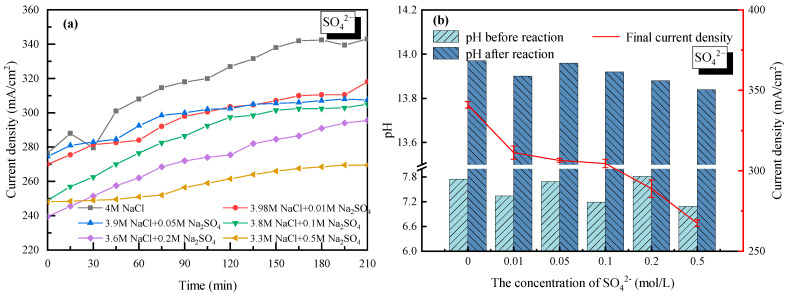
Effect of SO_4_^2−^ on the electrolysis performance of saline water. (**a**) Current density vs. time with SO_4_^2−^ doping; (**b**) pH before and after electrolysis with SO_4_^2−^ doping.

**Figure 8 molecules-28-04576-f008:**
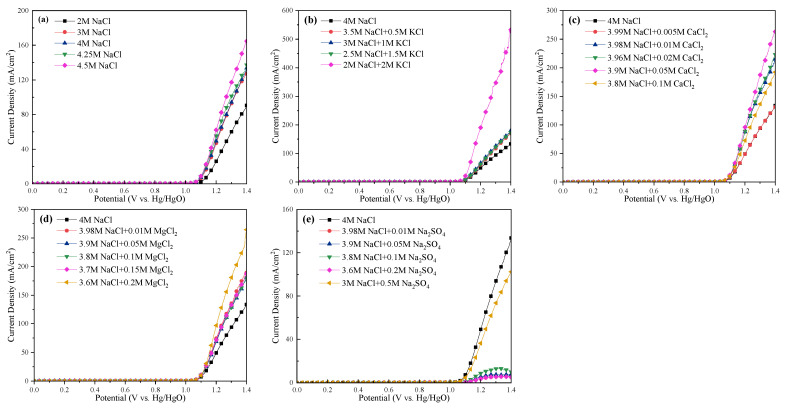
LSV curves of the anode at different electrolysis conditions. (**a**) Effect of NaCl concentration; (**b**) effect of K^+^; (**c**) effect of Ca^2+^; (**d**) effect of Mg^2+^; (**e**) effect of SO_4_^2−^.

**Figure 9 molecules-28-04576-f009:**
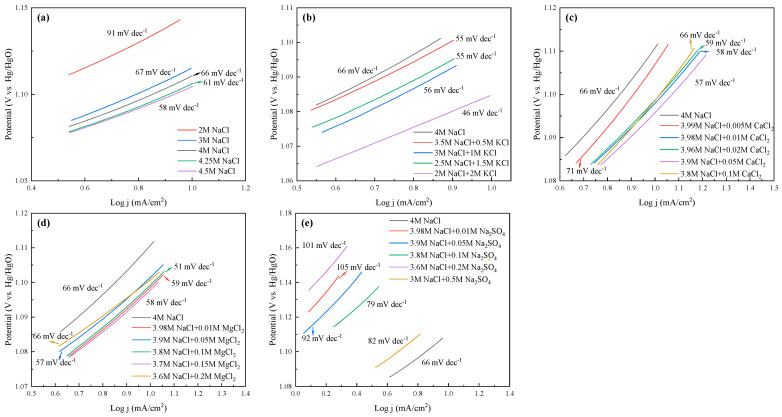
Tafel slopes of the anode at different electrolysis conditions. (**a**) Effect of NaCl concentration; (**b**) effect of K^+^; (**c**) effect of Ca^2+^; (**d**) effect of Mg^2+^; (**e**) effect of SO_4_^2−^.

**Figure 10 molecules-28-04576-f010:**
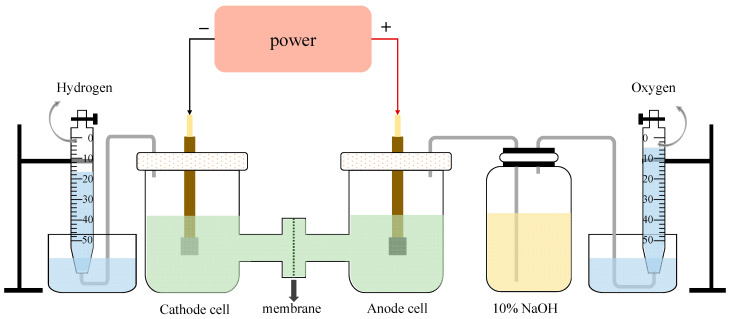
The schematic diagram of electrolysis equipment.

**Table 1 molecules-28-04576-t001:** Composition of the saltwater solutions containing impurity ions.

Group	Concentration of the Ions (mol/L)					
	Na^+^	K^+^	Ca^2+^	Mg^2+^	Cl^−^	SO_4_^2−^
Group 0 (Control group)	4	0	0	0	0	0
Group 1 (Effect of K^+^)	3.5	0.5	0	0	4	0
	3	1				
	2.5	1.5				
	2	2				
Group 2 (Effect of Ca^2+^)	3.99	0	0.005	0	4	0
	3.98		0.01			
	3.96		0.02			
	3.9		0.05			
	3.8		0.1			
Group 3 (Effect of Mg^2+^)	3.95	0	0	0.01	4	0
	3.9			0.05		
	3.8			0.1		
	3.7			0.15		
	3.6			0.2		
Group 4 (Effect of SO_4_^2−^)	4	0	0	0	3.98	0.01
					3.9	0.05
					3.8	0.1
					3.6	0.2
					3	0.5

## Data Availability

Data is contained within the article and Appendix A.

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
