# Peer review of "In-Depth Study on the Effects of Impurity Ions in Saline Wastewater Electrolysis"

_molecules, 2023, doi:10.3390/molecules28124576_

Round 1

Reviewer 1 Report

It is well known that in the chlor-alkali industry, when membrane electrolysis is used, the presence of calcium and magnesium is limited to a few ppb. So what sense does it make to test membrane electrolysis of saline solutions with a high concentration of ions of these elements. The "Introduction" chapter does not give an explanation for this.

The design of the electrolyzer shown in Figure 8 is far different from industrial realities.

The conditions under which the experiments were conducted, the results of which are presented in Figures 1-7, are not shown.

The manuscript contains many linguistic errors, incorrect terms as well as inaccuracies.

36-37

“With the widespread application of brine electrolysis technology in seawater desalination, … This is not true.”

41

“ion membrane” – improper term

142-144

“Meanwhile, the slope of Cl2 production was a little bit lower compared with H2, which was mainly due to the fact that a trace of O2 precipitation reaction also occurred at the anode at the same time.”– “precipitation” ? – should be “evolving”

240-242

“The cation exchange membrane was purchased from Nafion N117, a perfluoro sulfonic acid ion membrane manufactured by Dupont, U.S.A.” - syntax error

244-245

“Considering the competitive precipitation of O2 and Cl2 at the anode side” – “precipitation” – improper term

253 -255

“The electrode reactions occurring in the anode and cathode chambers were the chlorination and hydrogen precipitation reactions respectively.” “chlorination”, “precipitation” – improper terms

194-195

“As the electrolysis proceeded, part of OH- passed through the cation exchange membrane due to reverse osmosis.” - I see no reason to explain this phenomenon by reverse osmosis.

196-198

“Overall, the allowable concentration of Mg2+ and Ca2+ was 0.1 mol/L and 0.01 mol/L, indicating that the negative effect of Ca2+ was more significant than Mg2+.” - I find no justification for the values given.

216-217

“Overall, the concentration range of SO42- ≤ 0.1 mol/L was allowable for electrolysis.” - I find no justification for the values given.

Reviewer 2 Report

Dong et al. explored the salt concentration for stable dechlorination and talked in depth about the effect of typical ions such as K+, Ca2+, Mg2+, and SO42-. This work is a good contribution to the field and could be published in the molecules after major revision as mentioned below:

1. The authors have discussed the ions for K+, Ca2+, Mg2+ and SO42-, but lack an in-depth exploration of their principles. To improve the depth of the article, it needs to be discussed.

2. The SEM in Figure 6 of the article cannot show the effect of ions on the membrane, please test the change of the membrane before and after the reaction for SEM as well as the TEM, which can show the change of the reaction more visually.

3. For the collection of Cl ions from the anode, whether is it affected by oxygen? What methods are used by the authors to reduce this effect?

Reviewer 3 Report

The authors reported the effects of impurity ions in saline wastewater electrolysis. They found that K+ had a positive effect on the H2/Cl2 production of saline wastewater through accelerating the mass transfer efficiency in the electrolyte, while the existence of Ca2+ and Mg2+ had negative effects on the electrolysis performance by forming precipitates, which would adhere to the membrane to reduce the membrane permeability, occupy the active sites on the cathode surface, and also increase the transport resistance of the electrons in the 20 electrolyte. This work can be considered for acceptance after the following revisions.

1.      The detailed test conditions should be stated. What are the anode and cathode?

2.      They authors studied impurity ions in saline wastewater electrolysis. However, the main work is concerned on the effect of different ions on the electrolysis performance since impurity ions are usually at very low levels. In addition, there is no related wastewater. Therefore, the title should be corrected as “In-depth study on the effects of metal ions in saline water electrolysis”.

3.      Why is the trend of current density for 4.25 mol/L NaCl in Fig. 1 different from others? The authors are suggested to retest it.

4.      Please give the LSV curves and Tafel slopes for all the tests.

5.      The electrolysis mechanisms should be discussed. Relevant references in this aspect include but not limits to: Molecules, DOI: 10.3390/molecules26216326; Nano Res., DOI: 10.1007/s12274-022-4874-7; Molecules, DOI: 10.3390/molecules27041222; Chin. J. Catal., DOI: 10.1016/S1872-2067(21)64030-5; Molecules, DOI: 10.3390/molecules27217308.

Reviewer 4 Report

The manuscript deserves publications, however, major revisions are needed:

- all effect described here should be supported by polarization curves and Tafel analysis, then, these experiments and results should be reported.

- Na+, as other cations, blocks the membrane by specific mechanisms, not by charge attraction. Then, all mechanisms should be explained for all cations. Fig 8 is not clear and non-depth discussion was reported. 

- the distance between both electrodes influences on the cell E, then, different distances and ionic strength could be tested. 

- effect on the pH in all cases should be explained based on the active chlorine species and pourbaix diagram. 

- authoritative reviews related to the use of electrochemical technologies in production of active chlorine species were omitted. Fox example: 10.1016/j.apcatb.2023.122430 and others

- No enough experimental information related to the instrumental analysis performed and the measurements done, was reported here. Then, no reproducibility is possible. 

- For all experiments, current efficiency should be estimated according the reactions and electrons transfer. 

- More chemical/electrochemical reactions must be used to support the results presented here. For example: 10.1016/j.apcatb.2006.03.023 and others

Round 2

Reviewer 1 Report

Since the solubility of magnesium hydroxide is several orders of magnitude lower than that of calcium hydroxide, it is to be expected that the magnesium ion will cause more problems than the calcium ion. In my opinion, Fig. 4 does not provide a basis for the statements expressed by the authors and the correlations are random and are more likely due to the irrational design of the electrolyzer. I also consider the explanation of the influence of sulfate ions to have no basis.

Author Response

(1) Since the solubility of magnesium hydroxide is several orders of magnitude lower than that of calcium hydroxide, it is to be expected that the magnesium ion will cause more problems than the calcium ion.

R: It is an objective but wrong opinion. In fact, the influence of impurity ions on the ion membrane was not measured by solubility alone. Our repeated experiments showed that the Mg(OH)2 precipitated on the surface of the membrane was loose in texture and would fall off by itself after reaching a certain thickness, making the blocked part of the membrane exposed again. However, Ca(OH)2 precipitates on or inside the membrane was tight and hard to be removed by sweeping. In this case, the damage of Ca to the membrane was irreversible and even worse than Mg. (please refer to the last paragraph and also figure 6 in Section 2.3 of our manuscript).

(2) Fig. 4 does not provide a basis for the statements expressed by the authors and the correlations are random and are more likely due to the irrational design of the electrolyzer.

R: Sorry for the misunderstanding. Dots connected by solid line was applied to show the current density in those figures. Moreover, the figure lengends were updated to make the experiment conditions more clear.

(3) I also consider the explanation of the influence of sulfate ions to have no basis.

R: Related reference was added to provide evidence on the effects of sulfate ions in this work. Moreover, the LSV curves and Tafel slopes of all cases were added to analyze the effect of sulfate ions.

Reviewer 2 Report

The current manuscript can be accepted.

Author Response

Thanks.

Reviewer 3 Report

The manuscript can be accepted.

Author Response

Thanks.

Reviewer 4 Report

All concerns have bee addressed and the manuscript has been partially improved however, it seems that some authors (or groups) are excessively quoted but others barely or not at all cited. Even when an authoritative review was given as example in the first review round, the authors have omitted experts in the field of active chlorine and Cl2 evolution once again. 

Author Response

Point 1: All concerns have been addressed and the manuscript has been partially improved however, it seems that some authors (or groups) are excessively quoted but others barely or not at all cited. Even when an authoritative review was given as example in the first review round, the authors have omitted experts in the field of active chlorine and Cl2 evolution once again.

Response 1: Sorry for the omission. The following two authoritative papers were added in the revised manuscript (refer to Ref.[8] and [9]).

[8]    Martínez-Huitle CA, Rodrigo MA, Sirés I, Scialdone O. A critical review on latest innovations and future challenges of electrochemical technology for the abatement of organics in water[J]. Applied Catalysis B: Environmental. 2023, 328: 122430.

[9]    Vazquez-Gomez L, Ferro S, De Battisti A. Preparation and characterization of RuO2-IrO2-SnO2 ternary mixtures for advanced electrochemical technology[J]. Applied Catalysis B-Environmental. 2006, 67(1-2): 34-40.